# Social Outbreak in Chile, and Its Association with the Effects Biological, Psychological, Social, and Quality of Life

**DOI:** 10.3390/ijerph20237096

**Published:** 2023-11-22

**Authors:** Solange Parra-Soto, Samuel Duran-Aguero, Francisco Vargas-Silva, Katherine Vázquez-Morales, Rafael Pizarro-Mena

**Affiliations:** 1Departamento de Nutrición y Salud Pública, Universidad del Bío-Bío, Chillan 3780000, Chile; sparra@ubiobio.cl; 2School Cardiovascular and Metabolic Health, University of Glasgow, Glasgow G12 8QQ, UK; 3Facultad de Ciencias para el Cuidado de la Salud, Universidad San Sebastián, Sede Los Leones, Santiago 7500000, Chile; samuel.duran@uss.cl; 4Facultad de Odontología y Ciencias de la Rehabilitación, Universidad San Sebastián, Sede Los Leones, Santiago 7500000, Chile; franvargassilva@gmail.com (F.V.-S.); katherine.vasquez@uss.cl (K.V.-M.)

**Keywords:** social outbreak, biological effects, psychological effects, social effects, quality of life

## Abstract

The World Health Organization has defined collective violence as the instrumental use of violence by people who identify themselves as members of a group against other individuals and have political, economic, or social objectives. In Chile, the “Social Outbreak” was used to describe an episode of collective violence, which began on October 18, 2019, triggered by a multitude of socioeconomic and political factors, with protests and mobilizations in the country’s large and small cities; in central, commercial, and residential areas, that lasted for several months, affecting a large part of the population. The objective of the present study was to associate the social outbreak in Chile with its biological, psychological, and social effects on people’s health and quality of life, as well as its characteristics in terms of exposure, proximity, type, and frequency. This was a cross-sectional study with non-probabilistic national-level sampling, conducted from 28 November 2019, to 3 March 2020. The instrument had four sections. A total of 2651 participants answered the survey; 70.8% were female, and the mean age was 35.2. The main disturbances perceived were protests (70.9%), alarm sounds (68.1%), shooting sounds (59.0%), and tear gas bombs (56.9%). When quantifying the magnitude of these associations, people who had a medium exposure have a higher probability (OR: 1.99, CI: 1.58; 2.50) of suffering three or more biological effects than people that have a low exposure, while people with higher exposition have a 4.09 times higher probability (CI: 3.11; 5.38). A similar pattern was observed regarding psychological effects, although social effects were primarily experienced by those with high exposure. Social networks, TV, and radio were the most used media among people who perceived a greater effect. People who lived, worked, or shopped near the disturbance’s areas show a higher proportion negative effect.

## 1. Introduction

Social movements have become increasingly common worldwide in recent years [1]. Before the pandemic hit, they spread throughout in Hong Kong, Nicaragua, Chile, and Iran [2,3,4,5,6]. During the pandemic, similar movements emerged in Colombia, Peru, the United States, South Africa, Nigeria and Myanmar [7,8,9,10,11]. Only in Peru, more than one hundred social movements were reported in the first months of the pandemic [8]. This situation can be characterized as collective violence, according to the World Health Organization (WHO) [12]. Collective violence is defined as the instrumental use of violence by people who identify themselves as members of a group against other individuals, and have political, economic or social objectives. Some examples of this form of violence are wars, terrorism, uprisings and rebellions; organized aggression and extortion against prisoners and citizens; as well as fights between bands or gangs; violence derived from ethnic, religious or similar conflicts; and extortion by mafias [12,13].

In Chile, the “Social Outbreak” was used to describe an episode of collective violence [14]. The social outbreak in Chile started on 18 October 2019 in the capital city of the country. It was triggered by a multitude of factors, including socioeconomic and political conflicts, a crisis in political representation, economic vulnerabilities among middle-income sectors, and various issues related to the lack of policies addressing education, healthcare, and challenges, such as crime [15]. That day, widespread fare evasion took place in various subway stations, followed by the torching of several stations.

Social movements also spread to other cities [16]. These protest and riots were not confined to major urban centers and instead occurred simultaneously in both large and small cities throughout Chile. Although they were concentrated in the downtown areas, they also unfolded in more residential and trading areas. Therefore, a large part of the Chilean population had direct contact with the protests and its effects.

Subsequently, stringent control measures were imposed by public order institutions to manage the situation. Finally, social movements persisted to March 2020, declined with the arrival of the COVID-19 pandemic in Chile, and subsided for at least a year.

However, the high rate of vaccination at the national level (71.6%) [17] and the easing of quarantine restrictions in August 2021, generated the resurgence of the protests [18], which reflects that this problem is still latent. At present and for similar reasons, which have been going on for decades, there has been a social outbreak in France, Belgium, and Switzerland [19].

“Collateral victims” have been defined as those who would be the most affected by social movements because they live in the vicinity of the events, directly experiencing various manifestations of collective violence. The World Health Organization (WHO) analyze forms and contexts of violence, using an ecological approach that considers four levels: individual, relationship, community, and societal [13,20]. In the individual level, it is possible to describe biological factors regarding who receives the violence and who is a perpetrator. Relationship level defines how people establish diverse relationships and their influence on violent behaviors. Community level refers to the different settings in which social relationships occur and how they influence behavior. Finally, societal level is linked to the structure of society that encourages or inhibits violence and helps to maintain economic or social inequalities between groups in society [13].

Most of the significant studies concerning collective violence and its repercussions have primarily centered on contexts involving war, post-war periods, or terrorist attacks [20,21,22,23,24]. In contrast, the epidemiological study of these phenomena of distinctly social origin is still limited since their impact, considered to be lower intensity, has been far less frequently studied or has not been studied in the absence of armed conflicts, and therefore, unequally affects people across society [20].

It has been described that war times, post war, or natural disasters, lead to biopsychosocial issues and have an impact on the quality of life (QoL) of those who go through such events [25,26,27,28,29]. However, research on the impact of collective violence on health has been scarce and relatively disciplinary fragmented and has not considered all aspects of human beings. Most studies have focused on specific aspects of mental health, such as post-traumatic stress disorder (PTSD), depression, anxiety, sleep disorders, suicidal behavior, alcohol abuse, and the impact on QoL [4,13,20,21,30,31,32,33,34]. However, other dimensions of biopsychosocial health have received scant attention.

In line with the ecological model, collective violence has an impact on living conditions, brings about changes in the forms of production, causes employment precariousness, makes it difficult to satisfy basic human needs, increases poverty [35], and affects mental health. In a cohort study examining the longitudinal patterns and predictors of depressive symptoms trajectories before, during and after the Occupy Central/Umbrella Movement of Hong Kong in 2014, four depressive trajectories were identified: resistant (22.6% of the sample), resilient (37.0%), mild depressive symptoms (32.5%), and persistent moderate depression (8.0%). Baseline predictors that seemed to protect against persistent moderate depression included: higher household income, greater psychological resilience, more family harmony, higher family support, better self-rated health, and fewer depressive symptoms [36]. However, other aspects of people’s lives, such as the biological and social dimensions, have been minimally explored.

In the elderly population of Hong Kong, it has been observed that the scores of quality of life (QoL) indicators, which were used to assess the impact of age-friendly city policies and had been on the rise from 2009 to 2017, experienced a significant decline across all domains following the social outbreak and the COVID-19 pandemic [37].

Furthermore, it has been described that social unrest occurred in non-war contexts affects people to a different extent [20,38], and the greater the physical proximity, time of exposure and psychological proximity, the greater the impact [23]. Unlike wars or natural disasters, these events separately affect different areas of a city and groups of society, and change the environmental characteristics of the areas of residence. For this reason, it is important to investigate the features specific to collective violence arising from social outbreaks as a public health phenomenon: proximity, exposure, type, quantity, frequency, and intensity. Moreover, certain variables such as exposure to information channels (both digital and conventional), the risk of exposure to misinformation, and the amount of time dedicated to such sources have yet to be thoroughly explored or analyzed in studies pertaining to social outbreaks up to the present.

The inability of the states to address the diverse needs of the population, particularly in a post-pandemic scenario, is likely to result in a surge in social movements. These movements stem from a heightened sense of insecurity, health concerns, increasing food prices, economic downturns, job losses, income inequality, and various other factors. This underscores the importance of analyzing collective violence as a public health issue [23].

The literature has mostly analyzed the social movements in Hong Kong [32,39], however, a few studies in Latin America have described biopsychosocial effects of social movements on people.

The objective of the present study was to associate the social outbreak in Chile with the biological, psychological, and social effects on people’s health and quality of life, and its characteristics, in terms of exposure, proximity, type and frequency.

## 2. Materials and Methods

### 2.1. Study Design

Cross-sectional study.

### 2.2. Sample

A non-probabilistic sampling was carried out at a national level. Inclusion criteria were people aged 18 years or older living in continental or insular Chile with the ability to follow written instructions, without impaired vision (or that can be corrected with orthotics), and who were able to answer the self-reported online questionnaire. A Google Forms document was used; it was submitted using a national database (*n* = 5000) and distributed through social networks (Facebook, Instagram, and Twitter). The questionnaire was available from 28 November 2019 to 3 March 2020.

The participants, after accepting the Informed Consent, answered an online questionnaire titled “Characterization of the Biopsychosocial Effects on Health and Sleep of People after a Month of the Social Outbreak in Chile (2019)”. Upon accessing the Google Forms link, participants were required were required to start by reading and accepting the Informed Consent. If they chose to respond with a “No” to the Informed Consent, they were redirected to the end of the survey and thanked for their time without participating in the research. If they indicated “Yes” on the Informed Consent, the evaluation instrument was displayed for them to complete. The initial statement within the Informed Consent stipulated that being 18 years or older was a prerequisite for participation in the study. Additionally, the Informed Consent explicitly stated in other sections that it had to be answered individually, autonomously, and online. Furthermore, the title of the Informed Consent, along with the research’s objective explained within it, clearly indicated that this research pertained to Chile, acknowledging Chile’s unique status as both a continental and island nation.

The study was developed following the Declaration of Helsinki, taking into consideration the bioethical principles of research involving human beings and was submitted to the Ethics Committee of the Servicio de Salud Metropolitano Sur (South Metropolitan Health Service, number 99-25112019), who approved the study.

### 2.3. Assessment and Variables

The measuring instrument consisted of 35 questions distributed in 4 items. The instrument consisted of closed questions with a single answer, closed questions with multiple choices and questions with open answers.

The first section was aimed at collecting sociodemographic data (9 questions about sex, age, nationality, level of education, employment status, region of the country, and city of origin) and addressed collective violence (7 questions aimed at assessing proximity, exposure, type and frequency of the disturbances, the media people used to keep themselves informed, and how long they were exposed to such media). Regarding proximity, the questionnaire included the following as separate questions: how far they live, work, and shop from the areas of protests and disturbances. Specifically, in each question, proximity was determined based on well-known areas and the distance measured in meters, based on a five-level radius classification system; this has been previously described [6]. Next, the respondents that answered “far” to the three questions, were classified as “Far away from the disturbed area”. If the respondents answered, “2 to 3 blocks or less” in all three locations (home, work, or shopping), were classified as “Very near the disturbed area.” Finally, any score between these two extremes is classified as “Near the disturbed area”. In this sense, exposure was identified as high, medium and low.

The second item featured 3 questions aimed at assessing the effects of the social outbreak on the bio-psycho-social spheres, grouping a total of 35 bio-psycho-social symptoms (Yes/No). When it comes to assessing biological, psychological, and social symptoms, we have drawn upon the framework of comprehensive gerontological assessment as a reference point to identify these major aspects of individuals’ health [40]. Furthermore, we’ve taken into account a set of questions that serve as the initial basis for designing and identifying the symptoms used in this research. These questions were already part of the Adult Preventive Medicine Examination, covering measurements, medical history, risk of falls, network identification, addiction assessment, and routine annual exams [41]. In addition, we have integrated elements from the Preventive Medicine Examination for the Elderly, which includes evaluation tools like the Barthel Index, Yesavage Geriatric Depression Scale, Pfeffer Functional Activities Questionnaire, Extended Mini-Mental State Examination, and Functional Evaluation of the Elderly (EFAM-CHILE) [42]. These exams are routinely conducted on individuals within these two age groups as part of primary healthcare in Chile. We have also incorporated certain symptoms based on personal accounts from individuals living in the vicinity of Ground Zero during the Social Outbreak [43]. Their accounts revealed the impact on their bio-psycho-social health. It is important to note that our research team, which includes a researcher residing within the same riot-affected area (within a 125 m radius), offers valuable insights rooted in the health, social, and cultural context of the symptoms we have examined.

In addition, this item incorporated questions aimed at investigating the effects of the social outbreak on self-perception of QoL, understanding QoL as the self-perception that an individual has of their position in life in the context of the culture and value systems in which they live and in relation to their objectives, expectations, norms and concerns, which, in its multidimensional character, considers aspects of the physical health, psychological state, level of autonomy, social relationships, beliefs and the relationship with the salient characteristics of the environment [44]. The question used to evaluate the effect on QoL was *To what extent do you feel that the Social Outbreak has affected your quality of life?* and comprised five categories: not at all; little; somewhat; to a great extent; extremely. QoL was categorized as (1) Not affected at all/little, (2) Somewhat and (3) To a great extent/extremely. It was also dichotomized as (0) Not affected at all/little and (1) Affected, including categories 2 and 3.

The Insomnia Severity Index (ISI) instrument was incorporated in the third item, and the fourth item incorporated the Epworth Sleepiness Scale (ESS); the results of these two last items have been recently published [6].

### 2.4. Statistical Analysis

Continuous variables are presented as mean and standard deviation (SD). Categorical variables are presented as frequency and percentage. The Chi square test or Fisher’s exact test were used to assess qualitative variables, while multivariable logistic regression was used to determine the association between biological, psychological and social effects (these variables were dichotomized: 0 = less than three effects, and 1 = more than three effects), and for assessing exposure (low, medium, and high) and disturbance variables with QoL [6]. The final regression model was obtained by means of the stepwise procedure; variables with a probability of association of 0.10 were kept in the final model. Odds ratios were used to show the magnitude of the association, and the confidence interval (CI) was set at 95%. The significance level was set at a *p*-value lower than 0.05 for the tests. All analyses were performed with STATA 16.1 software (StataCorp, College Station, TX, USA).

## 3. Results

A total of 2651 participants answered the survey; 23.5% of them had higher exposure, and 14.4% had lower exposure. Most participants were women (70.8%), and the average age was 35.22 years old (SD 11.3); when comparing the most exposed with the least exposed, the most exposed participants were younger, mainly higher education students and dependent and independent workers (Table 1).

When asked if they have experienced biological, psychological and social effects, most participants reported psychological effects, followed by biological and finally social effects (Table 2 and Appendix A). When these effects were examined according to the level of exposure, for the different types of effects, the most exposed subjects had higher frequencies. In the case of the least exposed, the mean for effects was 8.07 (SD 5.81), while for the most exposed, it was 11.91 (SD 5.74) (Table 2).

When taking a close look at the biological effects, namely “have you felt more physically tired?”, “have your eyes become irritated?”, and “are you eating more?”, these three effects were the ones most affecting the studied population, and all of them were greater in the most exposed subjects (Appendix A).

Meanwhile, regarding psychological effects, “have you felt more mentally tired?”, “have you felt more stressed?”, “have you felt more anxious?”, and “have you felt more distressed?” were the most common effects. A similar trend is observed when comparing by exposure; the most exposed subjects have higher frequencies (Appendix A).

Finally, “have you experienced conflicts with your family or friends?”, “have you reduced your involvement in recreational or cultural groups in which you participated regularly?”, and “have you isolated or distanced from family and friends?” were the most frequent social effects, especially among the most exposed subjects (Appendix A).

When quantifying the magnitude of these associations, people who had a medium exposure had a 1.99-times higher probability (CI: 1.58; 2.50) of suffering three or more biological effects than people that have a low exposure, while people with higher exposure have a 4.09-times higher probability (CI: 3.11; 5.38). This was like the risk of experiencing psychological effects (Medium: OR: 1.76 (CI: 1.38; 2.50); High: OR: 3.23 (CI: 2.37; 4.40)); however, social effects occur mainly in people with high exposure (OR: 2.13 (CI: 1.55; 2.92), as seen in Figure 1.

Most of the participants who had their QoL affected “to a great extent” or “extremely” affected were women (32.5%), people with a college/technical education degree (38.5%), workers (42.7%), and people living in the Metropolitan Region (39.7%) (Appendix A).

When observing the association between the proximity to the disturbances and the place where they live, people who had their QoL affected were those who lived in the area of the disturbances (64.5% “to a great extent/extremely”; 23.7% “somewhat”), and those who lived 2 to 3 blocks around the disturbances (37.7% “to a great extent/extremely”; 40.1% “somewhat”), while most of the participants who lived more than 5 blocks away from the disturbances perceived their QoL “not affected at all” or “little” affected. The largest was the distance, and the lowest was the association. For instance, compared to people in areas without disturbance, people living within the disturbance place experienced a 5.72-times greater effect on their QoL (OR: 5.72, 95% CI: 3.54; 9.23). A similar trend was observed when the disturbances occurred near the workplace (OR: 2.92, 95% CI: 2.02; 4.22) and shopping place (OR: 3.65, 95% CI: 2.38; 5.61). People within the disturbed place experienced a greater effect on their QoL (Appendix A).

The main disturbances reported were protests (70.9%), alarm sounds (68.1%), shooting sounds (59.0%), and tear gas bombs (56.9%) (Appendix A). The frequency of disturbances was higher among people who felt their QoL largely affected. The types of disturbance which affected their QoL more were smoke (OR: 1.60, 95% CI: 1.28; 1.58), gunshot noises (OR: 1.43, 95% CI: 1.16; 1.77), and protests (OR: 1.42, 95% CI: 1.15; 1.75) (Table 3). People who perceived a greater effect on their QoL reported an average of 5.6 types of disturbances, while people who did not perceive a significant effect reported only 3.6 types (*p* value < 0.001). Regarding the frequency of the disturbances, participants with the most affected QoL pointed out that events occurred “almost every day of the week and every week in the last month” and “several days a week and more than three weeks in the last month” (60.3% and 34.7%, respectively), while for those who did not perceive their QoL as being affected, the figures were 11.3% and 21.2%, respectively. Increasing the frequency resulted in the highest effect on QoL, and when the frequency was almost every day, these people experienced an 11.4-times greater effect on their QoL compare to people who have not experienced these disturbances (OR: 11.40, 95% CI: 7.11; 18.3) (Table 3).

Regarding the question of how they keep up to date with national news events, the main media mentioned were social networks (Facebook, 67.3%; Instagram, 60.2%; WhatsApp, 48.4%; and Twitter, 41.6%), television (65.5%), and radio (49.5%) (Appendix A). WhatsApp (OR: 1.49, 95% CI: 1.24; 1.80), Twitter (OR: 1.40, 95% CI: 1.17; 1.23), and newspaper (OR: 1.38, 95% CI: 1.12; 1.69) showed the higher effect in their QoL; participants who spent more than 4 h checking media also had their QoL more affected (OR: 1.90, 95% CI: 1.51; 2.40) (Table 4).

## 4. Discussion

The main result of the present study is that people who had greater exposure to protests and riots during the social outbreak experience greater negative effects on the psychological, biological and social aspects (in that order), than those who do not perform day-to-day activities near the demonstrations. Likewise, the QoL was affected, which is mainly associated with factors such as proximity, exposure, type, number and frequency of disturbances, as well as the media used to keep themselves informed.

In Chile, the Social Outbreak persisted from October 2019 to March 2020 declined with the arrival of the COVID-19 pandemic in Chile and subsided for at least a year. By August 30, 2021, Chile had successfully vaccinated a total of 13,480,786 residents, representing 71.6% of the entire population [17]. As a result, the health authority substantially reduced confinement measures at the national level. Consequently, social movements resurfaced during August 2021 [18] in the so-called “Ground Zero” areas of the country [43].

Furthermore, to mark the two years anniversary of the social outbreak in Chile, on 18 October 2021 [45], protests once again took place in 12 out of 25 “Ground Zero” areas of the capital city. This resurgence of protests brought back the uncertainty in the population, potentially rekindling the negative bio-psycho-social effects previously mentioned in the present research.

As well as in Chile, several social movements have recently taken place in different countries around the world, such as Hong Kong, Lebanon, Iraq, Israel, Ecuador, Bolivia, Puerto Rico, Colombia, Ethiopia, Myanmar, Nicaragua, Nigeria, and South Africa [3,4,7,9,10], including during this year in France, Belgium and Switzerland [19]. These movements have raised the interest of analysts and decision makers who have made efforts to understand what may be the largest wave of mass social movements in world history [46]. One recurring aspect among all of these movements is the presence of collective violence.

Although there is controversy, some authors state that protests have been mostly peaceful. However, others suggest that violence has been a necessary strategy in conditions of outright repression [47], or an undesired consequence arising from the extent of the demonstrations and the difficulty of coordinating actions in a movement without visible leaders [48].

From a particular social perspective, in situations of collective violence, the establishment of an atmosphere of fear, anxiety, insecurity, hopelessness, and distrust of society and institutions has been described that seem to favor social relations marked by polarization and stereotyped beliefs [20]. It has been reported that large-scale, long-term population-related stress factors, such as social unrest, are associated with an increase in the symptoms of PTSD, general health problems [11], and mental health issues [4,5,33], such as depression [4,32] and insomnia [6,49], and disrupted QoL [37].

As demonstrated and identified with our research, the effects of the social outbreak negatively affect several spheres of life and QoL of people. Some of the most reported symptoms by the participants of the present study were physical fatigue, mental fatigue, stress, anxiety, distress, sleepiness, sleep problems, irritability, and intolerance. Many of these are considered part of the diagnostic criteria consistent with depressive symptoms, acute stress and sleep–wake disorders [50]. In future social outbreak situations, it will be crucial to conduct investigations using symptoms and/or structured, validated evaluation instruments online. These investigations should focus on identifying the presence of or diagnosing mental health issues, such as stress or depression, similar to our previous exploration and reporting on sleep disorders and insomnia using structured, validated instruments in the initial publication of this research [6].

Social movements may cause stress, increasing depressive symptoms, as observed in the Occupy Central Movement (Hong Kong) [32]. In this context, individuals who exhibited higher levels of depressive symptoms were found to be correlated with increased exposure to online news about the disturbances through social networks. [39] and the disruption of services. It has also been noted that media coverage in the aftermath of collective traumas can significantly propagate acute stress [51]; which agrees with our study regarding the effects on health, since people who were informed through social networks, television and radio, and those who spent more hours online, experienced a greater impact on their QoL. In Chile, for several months and on a daily basis, both formal and informal mass media showed the riots and the effects of collective violence, which could be associated with the presence of symptomatology in the participants of our study.

Moreover, it has been described that in relation to the Hong Kong protests, the frequency of comments containing terms related to mass protests was notably higher on days with significant protests compared to days without them. Additionally, the frequency of comments with both mass protest and psychological distress terms was also higher on days with protests, showing a time-lagged effect (responses on the following day) of protest terms on online forums but not on social networks. These findings suggest a positive association between offline protest activities and online psychological reactions [33].

All of this information is highly pertinent when it comes to educating the public about the potential consequences of using social networks during periods of social unrest. This is with a view of preventing mental health issues and disrupted QoL since these platforms can sometimes be vehicles for the spread of misinformation and fake news.

Among the biological effects, eye irritation is one of the most noticeable during the research. This may be related to the smoke of the barricades and/or tear gas bombs in the areas of protests and riots. Also, 56.9% of the participants reported exposure to tear gas during the last month, which might even cause other immediate or delayed problems. In that regard, in a cross-sectional, self-administered survey available online from 30 July 2020, to 20 August 2020 in which 2257 adult participants reported recent exposure to tear gas in Portland, Oregon (EE.UU.), 93.8% of the respondents reported they had experienced physical (93.7%) or psychological health issues (72.4%) immediately after (93.3%) or days following (86.1%) the exposure, and a slightly higher proportion experienced delayed head or gastrointestinal tract issues compared with immediate complaints [52]. Another apparent biological effect is increased food intake, which can lead to body weight gain and finally to obesity, which has a significant impact on the individual’s physical, mental, and social health, and causes negative effects, such as an increased risk of suffering other chronic diseases, thus leading to increased healthcare expenditures [53].

Disturbances associated with social movements cause the early, temporary, or definitive closing of many public and private services, facilities, or meeting places attended by the public. This affects daily social activities and, as a consequence, leads to the loss of routines such as the participation in community groups or practicing physical activity, as mentioned in the study, which could be very detrimental to more vulnerable groups such as older people [3,7,9].

Some factors analyzed in our study indicate that the presence of some effects has doubled, as in the case of eye irritation, eating schedule disturbances, bruxism, the presence of pain, reduced appetite, starting to take medication, alcohol consumption, increased alcohol consumption, relapse into tobacco use, slower daily behavior, sport dropout. Furthermore, in other cases, the effects have even tripled, such as the increased use of medication and the occurrence of falls. Most of the aforementioned are classified as biological effects in people with high exposure to protests, which might have repercussions in the QoL and could lead to mental health issues and other pathologies [54,55,56].

It has been described that Hong Kong protests led to disabilities and had an impact on physical and mental health, perceived stigma and social participation, with younger adults showing higher levels of political participation, symptoms of PTSD, functional disability and perceived stigma, which indicates that younger adults’ needs are different from those of older adults [57].

It has been described that the greater the physical proximity, time of exposure, and psychological proximity, the greater the impact of collective violence [23]. Thus, our results show the profile of those who had their QoL most affected by social outbreaks; the collateral victims, that is, people who lived, worked or went shopping less than 500 m away from the disturbances; people who experienced five or more types of disturbances; those who experienced disturbances on a daily basis, and every week during the last month; and those who checked four or more types of media to keep themselves informed for 4 h or more. In addition, the study provides relevant information for the development of preventive assessment and social-health interventions to avoid QoL being affected.

Worldwide leading health organizations have emphasized the importance of QoL and well-being as a goal across all life stages; furthermore, QoL is a predictor of survival [58]. QoL has been used to monitor the efficacy of health services, to assess intervention outcomes, and as an indicator of unmet needs [58]. In addition, multimorbidity has been described to be more prevalent among older people, with poor quality of life being one of the major consequences [59]. Therefore, given the evidence regarding the longitudinal relationship between QoL and mortality risk, the utility of a QoL assessment tool may improve health, thus reducing mortality [58]. For that reason, it is fundamental to emphasize QoL assessment as an indicator of the effects of collective violence resulting from social outbreaks in order to measure their impact on the general population and in particular groups, such as older people, so as to develop comprehensive interventions (promotion, prevention, treatment, and rehabilitation).

The WHO acknowledges the role of public health structures and other stakeholders in addressing violence, and urges them to take steps to tackle this problem by means of its characterization and assessment, and to adopt interventions aimed at the prevention of its effects on health [23].

However, an important aspect to take into account, and that we have not studied yet, is that social outbreaks cause supply chain breakdowns, and disrupt services and healthcare provision. These disruptions can potentially exacerbate the negative bio-psycho-social effects and further impact QoL. This is because social-health services may struggle to deliver timely responses to deal with the effects of social outbreaks, a situation that has been described in Myanmar and South Africa [7,9], where both patients and medical staff, as well as critical supplies, could not reach health facilities. Furthermore, medical staff were unable to work because of public transport disruptions, road closures, suspension of ambulance services, the looting of pharmacies and the shutdown of clinics, disrupting the process of vaccination against COVID-19 [9]. This means that in the face of future social unrest events, public and private healthcare services should learn from their experience or from foreign experience and adjust their policies accordingly.

In this way, non-governmental organizations (NGOs) that play a role in the provision of certain healthcare services, have experienced significant disruptions [3]. For this reason, it is important that these NGOs that cooperate with the formal healthcare system can generate an institutional plan of action, communication strategies, and partnerships that allow them to anticipate and mitigate the adverse impacts of social outbreaks [3].

The Community-Based Inclusive Development approach has been proposed to examine needs regarding the health, inclusion, and social participation of people during social unrest. It encourages the establishment of resilient, inclusive, and equitable communities, in which people are empowered and are given the opportunity to lead inclusive and healthy lives [57]. This approach might be useful to explore community health needs in geographical areas of Chile where there is a higher occurrence of disturbances, the so-called “Ground Zero” [43] areas in which people live, work or go shopping and that were found to affect their health and QoL, similar to if it were to happen in any area of a city where social outbreaks are generated.

This approach could also be used specifically and according to health problems, as was already carried out in an investigation with people with PTSD secondary to social outbreaks [60], where a conceptual model was created to describe and understand the state of health and holistic needs of these people based on the International Classification of Functioning, Disability, and Health (ICF) [61]. There were four essential areas: post-traumatic distress symptoms, participation restrictions, perceived stigma, and functional disability. Following the ICF framework, health condition, stigma, and participation restriction contributed crucially to disability and functioning, which allows for the highlighting of the importance of assessing the impacts on mental health, considering the interactions with rapid change and the stressful social environment [60].

From a global perspective, a model has been developed to attempt to explain how “big events”, such as social outbreaks, can affect a series of conditions related to society, life, subcultural norms, risk environments, practices, and risk and safety nets [62]; this model could contribute to explain how social outbreak have had an impact on people’s QoL and health globally in countries [6]. Therefore, dealing with this public health phenomenon poses great challenges in the fields of research, intervention, and teaching.

Our research demonstrates the effects of social outbreak were primarily psychological and mood-related. These effects were exacerbated by the COVID-19 pandemic, both in Chile and at the global level.

In the general population who endured lockdowns during the pandemic, studies have reported a high prevalence of distress symptoms and psychological disorders, specifically, general psychological symptoms, emotional disturbance, depression, stress, low morale, irritability, insomnia, symptoms of PTSD, anger, and emotional exhaustion. Bad mood and irritability stand out for their high prevalence [63].

It is important to mention that the Social Outbreak in Chile stopped with the arrival of the COVID-19 pandemic, and then was reignited at different times, namely when the best vaccination rates were shown or when there were decreases in quarantines imposed by the health authority [15,17,18]; there were even some points in time when both public health phenomena occurred simultaneously.

The COVID-19 pandemic has also modified people’s routines, caused disruptions in employment, and jeopardized the supply of food and essential services. It is likely that the COVID-19 pandemic, along with economic problems affecting the countries, causes an increase in collective violence [7] and social outbreaks at a global level. The pandemic has also exacerbated long-standing inequalities such as poverty, unemployment, and the lack of available resources, therefore increasing the risk of collective violence and worsening its consequences [9,10]. An example of the aforementioned is a study conducted in the United States during the pandemic, which showed that 25% of the respondents had symptoms suggestive of high levels of depression [64]. Countries should prepare their health systems for many people who will develop mental health problems, sleep disorders, and general health problems. That said, both phenomena, that is, social outbreaks and the COVID-19 pandemic, might co-occur and boost their harmful effects, affecting the bio-psycho-social health and QoL of people. There is already evidence of the impact of both phenomena on mental health in Hong Kong [65].

### 4.1. Implications for Future Practice

As has been pointed out, collective violence is a public health problem that should be addressed and prevented as it affects several spheres of the person and their QoL. To this date, there have been no public policies addressing social outbreak issues concerning collective violence, and consequently, there have been no substantial promotion or prevention strategies with isolated actions carried out by some non-governmental organizations (NGOs) being the only measures to be taken.

In this context, having a list of potential symptoms would allow for the implementation of a quick screening, conducted via telephone, e-mail, or social networks, which are low cost, simple, far-reaching, timely, and rapid assessment instruments, as reported in this research, including the surveys that were widely used during social unrest and the COVID-19 pandemic [66,67,68].

Therefore, online assessment tools should be explored to identify possible predictors and risk factors for PTSD, effects on health, and QoL, as a consequence of collective violence in order to promote timely detection, good predictability of potential health outcomes, and early intervention [2].

Likewise, and considering the present and future recurrence of social movements in several countries, it is recommended to encourage primary healthcare teams to generate strategies to be applied to the general population and to implement protective factors for bio-psycho-social health and QoL, against the constant probability of the occurrence of social outbreaks at any time.

In addition, at the governmental level, methodologies could be implemented that allow the use of data mining from social networks to identify the feelings of people in situations of social outbreak and the interventions at the level of public policy as a consequence. This would allow for the transparent government monitoring of the responses to public policies at a population level, using the experience of the methodology proposed in the research carried out on twitter in Panama, considering the existence of linguistic regionalisms of the Spanish language, among Latin American countries [69].

### 4.2. Implications for Future Research

Faced with other events of social outbreak, there is a need to investigate specified groups, such as older adults and people with disabilities, in order to determine if there is a higher or lower impact on their bio-psycho-social health and QoL, and if particular attention is required.

At the same time, the research can be complemented with qualitative studies that allow us to identify from people’s stories how Social Outbreaks affect people’s health in their various spheres in greater depth, considering local and cultural aspects.

Similarly, structured assessments and tools should be incorporated into QoL analyses. In the same way, other structured evaluation tools, which allow for an investigation in greater depth into the effects of social outbreaks on people’s mental health, such as depression and anxiety, which complement other aspects that have already been investigated, such as insomnia and daytime sleepiness [6].

To the extent that promotional, preventive, and therapeutic interventions are implemented to avoid the negative effects of social outbreak on individuals, families, and communities, they should be evaluated, as was already achieved in an investigation and intervention aimed at combating the psychospiritual and cognitive sequelae of social outbreaks and the COVID-19 pandemic in people with Parkinson’s disease [70].

### 4.3. Strengths and Limitations

Among the weaknesses, we can mention that this is a cross-sectional study, so we cannot speak of causality but only of association, and that there was a low proportion of surveys answered by men. In addition, since the survey was conducted online, people with lower incomes, lower educational levels, connectivity problems, the visually impaired and those who are not familiar with social networks, such as older adults, were excluded from the study. However, in future research face to face survey could improve the representativeness to our sample. On the other hand, Among the strengths of this study, we can highlight that nationwide coverage was achieved, which does not centralize the issues in the capital city. Also, the conceptualization of collective violence was expanded, with the analysis of its effects on people’s health and quality of life, secondary to social outbreaks (their manifestations and disturbances), contributing to its study as a phenomenon of public health.

## 5. Conclusions

People who frequently carry out day–day activities near the areas of social movements show a higher proportion of adverse effects on biological, psychological, social aspects, and QoL. Such effects were influenced by the proximity, frequency, and number of disturbances, as well as by mass media used by the participants to keep themselves informed.

It is important to recognize Social Outbreaks as a public health phenomenon, which is becoming increasingly prevalent due to globalization and the real-time flow of information via social networks. Similarly, it is crucial to incorporate this phenomenon into the curriculum for undergraduate and postgraduate education in public health for various professionals and disciplines, and to emphasize its impact on people’s health as a result of these Social Outbreaks.

It is necessary that healthcare teams, ministries, and governments identify the effects that social movements have on the community and people’s health in order to promptly address such effects and consider the implementation of a health screening to evaluate the bio-psycho-social and QoL effects on people facing collective violence due to social outbreaks. It is essential to further investigate this subject using structured, validated, and online evaluation tools to gather more data for future health management. Additionally, we should promote qualitative research that enables a comprehensive understanding, taking into account the perspectives of individuals and local cultural factors. This approach will help us to identify the diverse health effects experienced by people during Social Outbreaks, complementing the findings already presented in this research and providing a deeper insight into this public health phenomenon.

## Figures and Tables

**Figure 1 ijerph-20-07096-f001:**
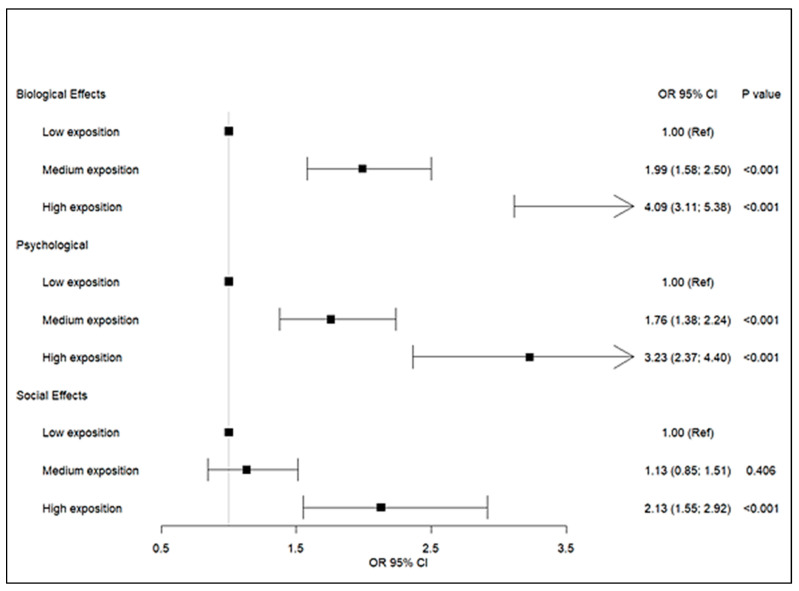
Association effects and exposition. Data are presented in odd ratio (OR) with 95% confidence intervals (95% CI). The reference group was people with low exposition. Model was adjusted for sex and region. Multivariate logistic regression was performed; *p* < 0.05.

**Table 1 ijerph-20-07096-t001:** Baseline characteristics by exposition.

	Overall	No Exposure	Medium Exposure	Higher Exposure	*p*-Value
**n**	2651	383 (14.4)	1645 (62.1)	623 (23.5)	
**Sex (%)**					
Man	774 (29.2)	105 (13.6)	485 (62.7)	184 (23.7)	0.709
Woman	1877 (70.8)	278 (14.8)	1160 (61.8)	439 (23.4)	
**Age (mean (SD))**	35.2 (11.30)	39.72 (14.82)	34.39 (10.23)	34.66 (10.88)	<0.001
**Age category (%)**					
30 or less	1125 (42.4)	137 (12.2)	711 (63.2)	277 (24.6)	<0.001
31 to 40	861 (32.5)	103 (12.0)	556 (64.6)	202 (23.4)	
41 to 50	370 (14.0)	61 (16.5)	231 (62.4)	78 (21.1)	
51 or more	295 (11.1)	82 (27.8)	147 (49.8)	66 (22.4)	
**Education (%)**					
Up to high school	96 (3.6)	16 (16.7)	58 (60.4)	22 (22.9)	0.006
Incomplete college	108 (4.1)	18 (16.7)	62 (57.4)	28 (25.9)	
College/Technical student	235 (8.9)	21 (8.9)	136 (57.9)	78 (33.2)	
College/Technical degree	2212 (83.4)	328 (14.8)	1389 (62.8)	495 (22.4)	
**Occupation (%)**					
Inactive	155 (5.8)	110 (71.0)	31 (20.0)	14 (9.0)	<0.001
Employer	157 (5.9)	21 (13.4)	94 (59.9)	42 (26.7)	
Student	315 (11.9)	26 (8.3)	190 (60.3)	99 (31.4)	
Worker	1462 (55.2)	155 (10.6)	977 (66.8)	330 (22.6)	
Independent	562 (21.2)	71 (12.6)	353 (62.8)	138 (24.7)	
**Zone (%)**					
North	229 (8.6)	24 (10.5)	140 (61.1)	65 (28.4)	0.051
Center	716 (27.0)	112 (15.6)	427 (59.6)	177 (24.8)	
South	280 (10.6)	45 (16.1)	186 (66.4)	49 (17.5)	
Metropolitan region	1426 (53.8)	202 (14.2)	892 (62.6)	332 (23.2	

Chi square test or Fisher’s exact test were used; *p* < 0.05.

**Table 2 ijerph-20-07096-t002:** Association effects and exposition.

	Overall	No Exposure	Medium Exposure	Higher Exposure	*p*-Value
**Suffer biologic problems (%)**					
Yes	2343 (88.4)	303 (12.9)	1462 (62.4)	578 (24.7)	<0.001
**Suffer psychological problems (%)**					
Yes	2532 (95.6)	349 (13.8)	1579 (62.4)	604 (23.8)	<0.001
**Suffer social problems (%)**					
Yes	2291 (86.4)	300 (13.1)	1424 (62.2)	567 (24.7)	<0.001
**Total negative effects (mean (SD))**	9.96 (5.69)	8.07 (5.81)	9.67 (5.43)	11.91 (5.74)	<0.001
**Affected QoL (%)**					
Not affected at all	75 (2.8)	20 (26.7)	44 (58.7)	11 (14.6)	<0.001
A little	715 (27.0)	146 (20.4)	450 (62.9)	119 (16.7)	
Somewhat	1064 (40.1)	118 (11.1)	719 (67.6)	227 (21.3)	
To a great extent	487 (18.4)	61 (12.5)	281 (57.7)	145 (29.8)	
Extremely	310 (11.7)	38 (12.3)	151 (48.7)	121 (39.0)	

Data presented as frequency with their percentage, only showing people who answered “yes”. Frequency and percentage, 100% based in “yes”. Chi square test were used to assess qualitative variables; *p* < 0.05.

**Table 3 ijerph-20-07096-t003:** Association disturbs type, quantity, and effect in QoL.

	Overall	Not Affected at All/Little	Somewhat	To a Great Extent/Extremely	OR 95% CI	*p*-Value
**Types of disturbance (%) ****						
Protests (%) *	1880 (70.9)	471 (25.1)	783 (41.6)	626 (33.3)	1.42 (1.15; 1.75)	0.001
Alarms/Horns of police, firefighters, or ambulances due to protests (%) *	1805 (68.1)	451 (25.0)	725 (40.2)	629 (34.8)	1.08 (0.88; 1.34)	0.457
Gunshot noises, explosions from homemade bombs, flares, or similar (%) *	1564 (59.0)	350 (22.4)	639 (40.9)	575 (36.7)	1.43 (1.16; 1.77)	0.001
Tear gas bombs (%) *	1508 (56.9)	349 (23.1)	616 (40.8)	543 (36.1)	1.35 (1.09; 1.66)	0.005
Barricade (%) *	1479 (55.8)	370 (25.0)	592 (40.0)	517 (35.0)	0.90 (0.73; 1.11)	0.328
Helicopter Noise (%) *	1441 (54.4)	342 (23.7)	594 (41.2)	505 (35.1)	1.30 (1.06; 1.58)	0.011
Smoke (%) *	1286 (48.5)	259 (20.1)	527 (41.0)	500 (38.9)	1.60 (1.28; 2.01)	<0.001
Looting (%) *	645 (24.3)	121 (18.8)	226 (35.0)	298 (46.2)	1.34 (1.04; 1.72)	0.026
Fire (%) *	626 (23.6)	110 (17.6)	223 (35.6)	293 (46.8)	1.30 (0.99; 1.70)	0.063
Neither (%) *	131 (4.9)	77 (58.8)	37 (28.2)	17 (13.0)	0.75 (0.49; 1.14)	0.182
**Quantity of Disturbance Events (mean (SD))**	4.61 (2.59)	3.57 (25.8)	4.63 (33.5)	5.63 (40.7)	1.30 (1.25; 1.35)	<0.001
**Disturbance Frequency (%)**						
Almost every day of the week and every week in the last month	426 (16.1)	48 (11.3)	121 (28.4)	257 (60.3)	11.40 (7.11; 18.30)	<0.001
Several days a week, and for more than 3 weeks in the last month	796 (30.0)	169 (21.2)	351 (44.1)	276 (34.7)	5.42 (3.63; 8.09)	<0.001
Only a couple of days (alternate or different) and only in 1 or 2 weeks in the last month	721 (27.2)	241 (33.4)	322 (44.7)	158 (21.9)	2.95 (1.99; 4.38)	<0.001
Only a couple of hours, and in isolation in this last month	582 (22.0)	260 (44.7)	232 (39.9)	90 (15.4)	1.80 (1.21; 2.68)	0.004
Never	126 (4.7)	72 (57.1)	38 (30.2)	16 (12.7)	1.00 (Ref.)	

* Only yes, Data presented as frequency and percentage, and odds ratio (OR) with their 95% confidence interval (CI). Model adjusted by sex, age, region, education and occupation. ** mutually adjusted. Multivariate logistic regression was performed; *p* < 0.05.

**Table 4 ijerph-20-07096-t004:** Association information media and effect in QoL.

	Overall	Not Affected at All/Little	Somewhat	To a Great Extent/Extremely	OR 95% CI	*p*-Value
**Media type for information (%) ****						
Facebook (%) *	1783 (67.3)	547 (30.7)	706 (39.6)	530 (29.7)	0.85 (0.70; 1.03)	0.105
TV (%) *	1736 (65.5)	490 (28.2)	723 (41.6)	523 (30.2)	1.07 (0.89; 1.29)	0.443
Instagram (%) *	1595 (60.2)	480 (30.1)	658 (41.3)	457 (28.6)	1.06 (0.86; 1.29)	0.589
Radio (%) *	1312 (49.5)	357 (27.2)	536 (40.9)	419 (31.9)	1.20 (1.00; 1.43)	0.045
WhatsApp (%) *	1283 (48.4)	320 (24.9)	510 (39.8)	453 (35.3)	1.49 (1.24; 1.80)	<0.001
Twitter (%) *	1104 (41.6)	283 (25.6)	445 (40.3)	376 (34.1)	1.40 (1.17; 1.68)	<0.001
People (%) *	920 (34.7)	251 (27.3)	377 (41.0)	292 (31.7)	1.05 (0.86; 1.27)	0.637
Newspaper (%) *	739 (27.9)	173 (23.4)	302 (40.9)	264 (35.7)	1.38 (1.12; 1.69)	0.002
**Number source information (mean (SD))**	3.95 (1.68)	3.44 (30.9)	3.7 (33.2)	4 (35.9)	1.17 (1.11; 1.23)	<0.001
**Hours dedicated to getting informed (%) ***						
0–1 h	664 (25.0)	257 (38.7)	241 (36.3)	166 (25.0)	1.00 (Ref.)	
2 h	778 (29.3)	233 (29.9)	339 (43.6)	206 (26.5)	1.54 (1.23; 1.93)	0.000
3 h	436 (16.4)	108 (24.8)	200 (45.9)	128 (29.3)	1.94 (1.47; 2.54)	0.000
4 or more h	773 (29.3)	192 (24.8)	284 (36.7)	297 (38.5)	1.90 (1.51; 2.40)	0.000

* Only yes. Data presented as frequency and percentage, and odds ratio (OR) with their 95% confidence interval (CI). Model adjusted by sex, age, region, education, and occupation. ** Mutually adjusted. Multivariate logistic regression was performed; *p* < 0.05.

## Data Availability

Data are available upon request.

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
