# Peer review of "Social Outbreak in Chile, and Its Association with the Effects Biological, Psychological, Social, and Quality of Life"

_ijerph, 2023, doi:10.3390/ijerph20237096_

Round 1
Reviewer 1 Report
Comments and Suggestions for Authors
Dear Authors.
You present a well thought out and presented article.
I have some doubts about your work in relation to the ethical code. You say: - "The study was developed following the Declaration of Helsinki, taking into consideration the bioethical principles of research involving human beings and was submitted to the Ethics Committee of the Servicio de Salud Metropolitano Sur (South Metropolitan Health Service), which approved the study". For me, that is not enough, I think there should be an approval of the Ethics Committee that guides the research, a numbering. It seems that the ethical criteria are followed but there is no approval from the ethics committee.
Another important doubt: how were the participants collected? How did they have guarantees that the inclusion criteria were followed? This raises many doubts for me to be able to accept your work. It is not clear to me, or at least it is not indicated in the methodology ("A Google Forms document was used and submitted using a national database (n=5,000), and distributed through social networks (Facebook, Instagram and Twitter). The questionnaire was available from 28th November 2019 to 3rd March 2020. ").
The results are clear and well presented.
I think the results are very interesting but not really connected to the implications it may have for day to day life. For example here: "As demonstrated and identified with our research, the effects of the social outbreak negatively affect several spheres of life and QoL of people. Some of the most commonly reported symptoms by the participants of the present study are: physical fatigue, mental fatigue, stress, anxiety, distress, sleepiness, sleep problems, feeling irritable and intolerant. Many of these are considered part of the diagnostic criteria consistent with depressive symptoms, acute stress and sleep wake disorders [46]."
Undoubtedly, social outbreak, they are potentially negative, but what really are their social implications on a day-to-day basis: work, family, etc.? I think they could be further connected to international references, and they are not. For example, this very important paragraph has only one reference.
I also do not understand at all the reference to Covid-19 (page 13). If it is not a studied element and the study is previous, why do they make this connection? I think it is inappropriate and takes the reader away from the progress of your article.
I find the conclusions very scarce. I believe that they need further development and a more decisive commitment from the socio-anthropological field on the part of the authors.
I believe that they can substantially improve their work. I encourage you to do so.
Author Response
Dear Reviewer, thank you very much for your comments, as they are very valuable for the improvement of our scientific article.
You present a well thought out and presented article.
I have some doubts about your work in relation to the ethical code. You say: - "The study was developed following the Declaration of Helsinki, taking into consideration the bioethical principles of research involving human beings and was submitted to the Ethics Committee of the Servicio de Salud Metropolitano Sur (South Metropolitan Health Service), which approved the study". For me, that is not enough, I think there should be an approval of the Ethics Committee that guides the research, a numbering. It seems that the ethical criteria are followed but there is no approval from the ethics committee.
Response: The ethics committee approval number was added (99-25112019). The study approval letter has been sent to the editor.
Another important doubt: how were the participants collected? How did they have guarantees that the inclusion criteria were followed? This raises many doubts for me to be able to accept your work. It is not clear to me, or at least it is not indicated in the methodology ("A Google Forms document was used and submitted using a national database (n=5,000), and distributed through social networks (Facebook, Instagram and Twitter). The questionnaire was available from 28th November 2019 to 3rd March 2020. ").
Response: The following sentence was added to the article:
Upon accessing the Google Forms link, participants were required were required to start by reading and accepting the Informed Consent. If they chose to respond with a "No" to the Informed Consent, they were redirected to the end of the survey and thanked for their time without participating in the research; In the event that they indicate “Yes” to the Informed Consent, the evaluation instrument was displayed for them to complete. The initial statement within the Informed Consent stipulated that being 18 years or older was a prerequisite for participation in the study. Additionally, the Informed Consent explicitly stated in other sections that it had to be answered individually, autonomously, and online. Furthermore, the title of the Informed Consent, along with the research's objective explained within it, clearly indicated that this research pertained to Chile, acknowledging Chile's unique status as both a continental and island nation.
With this, we ensured that the participants met the inclusion requirements of the research.
The results are clear and well presented.
Response: Thank you.
I think the results are very interesting but not really connected to the implications it may have for day to day life. For example here: "As demonstrated and identified with our research, the effects of the social outbreak negatively affect several spheres of life and QoL of people. Some of the most commonly reported symptoms by the participants of the present study are: physical fatigue, mental fatigue, stress, anxiety, distress, sleepiness, sleep problems, feeling irritable and intolerant. Many of these are considered part of the diagnostic criteria consistent with depressive symptoms, acute stress and sleep wake disorders [46]."
Response: The following sentence was added to the article:
In future social outbreak situations, it will be crucial to conduct investigations using symptoms and/or structured, validated evaluation instruments online. These investigations should focus on identifying the presence or diagnosing mental health issues such as stress or depression, similar to our previous exploration and reporting on sleep disorders and insomnia using structured, validated instruments in the initial publication of this research.
Undoubtedly, social outbreak, they are potentially negative, but what really are their social implications on a day-to-day basis: work, family, etc.? I think they could be further connected to international references, and they are not. For example, this very important paragraph has only one reference.
Response: Three references were added to reinforce and substantiate what was discussed in relation to the social effects identified in this research. Those detailed below:
- Hartmann, C., Hartmann, J., Lopez, A., Flores, P.: Health non-governmental organizations (NGOs) amidst civil unrest: Lessons learned from Nicaragua. Glob. Public Health. 15, 1810–1819 (2020). https://doi.org/10.1080/17441692.2020.1789193
- Aung, M., Shiu, C., Chen, W.: Amid political and civil unrest in Myanmar, health services are inaccessible. Lancet (London, England). 397, 1446 (2021). https://doi.org/10.1016/S0140-6736(21)00780-7
- Makoni, M.: Social unrest disrupts South African health care. Lancet (London, England). 398, 287 (2021). https://doi.org/10.1016/S0140-6736(21)01654-8
I also do not understand at all the reference to Covid-19 (page 13). If it is not a studied element and the study is previous, why do they make this connection? I think it is inappropriate and takes the reader away from the progress of your article.
Response: 2 sentences were mobilized for the end of the discussion, to better link what concerns COVID-19, highlighted in yellow.
Additionally, the following sentence is added to the article:
It is important to mention that the Social Outbreak in Chile stopped with the arrival of the COVID-19 pandemic, and then was reactivated at different times with the best vaccination rates and decreases in quarantines imposed by the health authority, which generated that even at some point both public health phenomena occurred simultaneously.
In addition, the following bibliographies are added to this phrase:
- Waissbluth, M.: Origins and evolution of the Social Outbreak in Chile, https://www.mariowaissbluth.com/descargas/mario_waissbluth_el_estallido_social_en_chile_v1_feb1.pdf, (2020)
- Mathieu, E., Ritchie, H., Rodés, L., Appel, C., Giattino, C., Hasell, J., Macdonald, B., Dattani, S., Beltekian, D., Ortiz, E., Roser, M.: Coronavirus (COVID-19) Vaccinations, https://ourworldindata.org/covid-vaccinations?country=OWID_WRL
- El mostrador: Demonstrations in Plaza Baquedano: attendees ask for pardon for prisoners of the outbreak, https://www.elmostrador.cl/dia/2021/08/06/manifestaciones-en-plaza-baquedano-asistentes-piden-indulto-para-presos-del-estallido/
I find the conclusions very scarce. I believe that they need further development and a more decisive commitment from the socio-anthropological field on the part of the authors.
Response: The following sentence was added to the article:
It is important to recognize Social Outbursts as a public health phenomenon, which is becoming increasingly prevalent due to globalization and the real-time flow of information via social networks. Similarly, it is crucial to incorporate this phenomenon into the curriculum for undergraduate and postgraduate education in public health for various professionals and disciplines, and to emphasize its impact on people's health as a result of these social outbursts.
It's essential to further investigate this subject using structured, validated, and online evaluation tools to gather more data for future health management. Additionally, we should advance in qualitative research that enables a comprehensive understanding, taking into account the perspectives of individuals and local cultural factors. This approach will help us identify the diverse health effects experienced by people during Social Outbreaks, complementing the findings already presented in this research and providing a deeper insight into this public health phenomenon.
I believe that they can substantially improve their work. I encourage you to do so.
Thank you so much, we appreciate your comments to improve our work.

Reviewer 2 Report
Comments and Suggestions for Authors
The article does explore an underexplored and important topic.
There are different concerns however with the article in its current form.
The QoL information is not sufficient. What was the exact wording of the QoL question? Was there only one question? Why was not a QoL established instrument used? How is QoL to be understood here?
In the supplemental material there appear to be questions that may be related to QoL dimensions. These should be examined as part of expanding the validity of the QoL claims being made.
It is difficult to understand in what ways the survey questions, except where specific information is provided, are related to or draw from established meausres/instruments for the different dimensions included. Further information is therefore needed. If adaptations have been made to better suit the cultural context, this information is also needed.
Additional cultural information is needed also to understand the inclusion of separate questions related to faith and spirituality, as well as for other variables that reflect the cultural context.
A reworking of the text is in line with the need for additional information, expected additional analysis, and further information relating to the cultural context.
Comments on the Quality of English Language
The quality of the English language necessitates a language review of the entire manuscript.
Author Response
Dear Reviewer, thank you very much for your comments, as they are very valuable for the improvement of our scientific article.
The article does explore an underexplored and important topic.
There are different concerns however with the article in its current form.
The QoL information is not sufficient. What was the exact wording of the QoL question? Was there only one question? Why was not a QoL established instrument used? How is QoL to be understood here?
In the supplemental material there appear to be questions that may be related to QoL dimensions. These should be examined as part of expanding the validity of the QoL claims being made.
Response: Within the methodology, the question was added with which the Quality of Life was consulted: To what extent do you feel that the Social Outbreak has affected your quality of life?
In addition, the concept of Quality of Life was added: Self-perception that an individual has of his position in life, in the context of the culture and value systems in which he lives, and in relation to his objectives, expectations, norms and concerns, which in its multidimensional character considers aspects of the physical health, psychological state, level of autonomy, social relationships, beliefs and the relationship with the salient characteristics of the environment.
The following bibliography was also added:
- Freire, N., Cabral, N., Marchioni, D., Vieira, S., De Oliveira, C.: Quality of life assessment instruments for adults: a systematic review of population-based studies. Health Qual. Life Outcomes. 18, 1–13 (2020). https://doi.org/10.1186/S12955-020-01347-7/FIGURES/4
The reason we didn't include a structured evaluation instrument is that we couldn't find one that was suitable for assessing Quality of Life in the context of the Social Outburst phenomenon we aimed to study. Additionally, it was uncertain how long this phenomenon would last. We were working within tight time constraints due to the project's formulation, presentation, and approval by the ethics committee. As a result, we prioritized a single question that could provide a comprehensive assessment of Quality of Life, with a focus on the specific aspects related to the phenomenon under investigation. It's worth noting that the Social Outbreak began on October 18, and we initiated the application of the evaluation instrument on November 28. Therefore, we believe that our research started investigating this phenomenon relatively quickly, and this represents a unique contribution to the existing body of knowledge up to this date.
It is difficult to understand in what ways the survey questions, except where specific information is provided, are related to or draw from established meausres/instruments for the different dimensions included. Further information is therefore needed. If adaptations have been made to better suit the cultural context, this information is also needed.
Additional cultural information is needed also to understand the inclusion of separate questions related to faith and spirituality, as well as for other variables that reflect the cultural context.
A reworking of the text is in line with the need for additional information, expected additional analysis, and further information relating to the cultural context.
Response: The following sentence is added to the article:
When it comes to assessing biological, psychological, and social symptoms, we've drawn upon the framework of comprehensive gerontological assessment as a reference point to identify these major aspects of individuals' health. Furthermore, we've taken into account a set of questions that serve as the initial basis for designing and identifying the symptoms used in this research. These questions were already part of the Adult Preventive Medicine Examination, covering measurements, medical history, risk of falls, network identification, addiction assessment, and routine annual exams.
In addition, we've integrated elements from the Preventive Medicine Examination for the Elderly, which includes evaluation tools like the Barthel Index, Yesavage Geriatric Depression Scale, Pfeffer Functional Activities Questionnaire, Extended Mini-Mental State Examination, and Functional Evaluation of the Elderly (EFAM-CHILE). These exams are routinely conducted on individuals within these two age groups as part of primary health care in Chile.
We've also incorporated certain symptoms based on personal accounts from individuals living in the vicinity of Ground Zero during the Social Outbreak. Their accounts revealed the impact on their bio-psychosocial health. It's important to note that our research team, which includes a researcher residing within the same riot-affected area (within a 125-meter radius), offers valuable insights rooted in the health, social, and cultural context of the symptoms we've examined. The following bibliographies are also added:
- Pizarro-Mena, R., Duran, S., Parra, S., Vargas, F., Bello, S., Fuentes, M.: Effects of a Structured Multicomponent Physical Exercise Intervention on Quality of Life and Biopsychosocial Health among Chilean Older Adults from the Community with Controlled Multimorbidity: A Pre-Post Design. Int. J. Environ. Res. Public Health. 19, (2022). https://doi.org/10.3390/IJERPH192315842
- Ministerio de Salud: AUGE Clinical Guide. Preventive Medicine Exam, (2013)
- Ministerio de Salud.: Technical Guidance for the health care of older adults in Primary Care, (2014)
- Poduje. I: Seven Heads. Urban chronicle of the social outbreak. Uqbar Editores (2020)
The quality of the English language necessitates a language review of the entire manuscript.
Response: The article has been reviewed by the one native speaker. Some adjustments have been made for better understanding, written in red.
Thank you so much, we appreciate your comments to improve our work.

Reviewer 3 Report
Comments and Suggestions for Authors
Some changes and improvements are needed:
Ln. 118 2.1. Study Design
Cross-sectional analytical study. Omit analytical, just cross-sectional study.
When discussing the results, refer in the discussion a little more to the sample that is an opportunity and does not include parts of the population that could not fill out the online questionnaire.
Author Response
Dear Reviewer, thank you very much for your comments, as they are very valuable for the improvement of our scientific article.
Some changes and improvements are needed:
Ln. 118 2.1. Study Design
Cross-sectional analytical study. Omit analytical, just cross-sectional study.
Response: This was amended
When discussing the results, refer in the discussion a little more to the sample that is an opportunity and does not include parts of the population that could not fill out the online questionnaire.
Response: The following sentence is added to the article:
The research can be complemented with qualitative studies that allow us to identify from people's stories, and in greater depth, how Social Outbreaks affect people's health in their various spheres, considering local and cultural aspects.
In future research face to face survey could improve the representativeness to our sample.
Thank you so much, we appreciate your comments to improve our work.
